# Topical 5% Imiquimod for the Treatment of Superficial and Nodular Periocular Basal Cell Carcinoma: A Systematic Review of Clinical Outcomes, Safety, and Treatment Strategies

**DOI:** 10.3390/cancers17132111

**Published:** 2025-06-24

**Authors:** Larysa Krajewska-Węglewicz, Piotr Sobolewski, Irena Walecka

**Affiliations:** 1Department of Ophthalmology, National Institute of Medicine of the Ministry of Interior and Administration, 02-507 Warsaw, Poland; larysa.krajewska@pimmswia.gov.pl; 2Dermatology Clinic, National Institute of Medicine of the Ministry of Interior and Administration, 02-507 Warsaw, Poland; irena.walecka@pimmswia.gov.pl

**Keywords:** imiquimod, basal cell carcinoma, topical treatment, outcomes, eyelid

## Abstract

Basal cell carcinoma is a common skin cancer affecting the eyelid, typically treated with surgery or radiotherapy. However, these methods can cause visible scarring or may not be suitable for some patients. This systematic review evaluated the use of imiquimod five percent cream as a non-invasive treatment option. The findings showed that the cream was effective in eliminating tumors in most cases, especially when the lesions were small and superficial. Patients generally experienced mild-to-moderate skin irritation around the eye, such as redness or crusting, but serious complications were not reported. Cosmetic outcomes were often better than those seen with surgery or radiation, with minimal scarring and preserved eyelid function. This treatment may be particularly helpful for individuals who are not good candidates for surgery or who prioritize appearance. The results support the use of imiquimod in selected cases and highlight the need for further studies on long-term outcomes and optimal treatment regimens.

## 1. Introduction

Eyelid tumors represent a clinically significant subset of cutaneous neoplasms due to their potential for local invasion, aesthetic deformity, and functional impairment. Among these, basal cell carcinoma (BCC) is by far the most prevalent, comprising over 90% of malignant eyelid tumors [1]. Its incidence continues to rise globally, attributed to aging populations and cumulative ultraviolet exposure, particularly among fair-skinned individuals [2,3]. UV exposure is the most significant risk factor for BCC, and accordingly the eyelid is a frequently involved site along with other sun-exposed areas (nose, lips, cheeks, forehead, hands, etc.) [4]. Though BCC metastasizes rarely, its potential for aggressive local invasion—especially in the anatomically sensitive periocular region—poses considerable challenges for both function and cosmesis [5].

The standard of periocular BCC management has traditionally involved surgical excision, often with Mohs micrographic surgery to ensure margin control and minimize recurrence [6]. Cure rates for surgical modalities typically exceed 95% [7]. However, surgery is not suitable for all patients, especially those of advanced age, with significant comorbidities, or with tumors located in cosmetically or functionally sensitive regions such as the medial canthus or lower eyelid margin [8]. In such cases, there is a clear clinical need for non-invasive, tissue-sparing treatment alternatives. Thus, in cases of treatment failure or contraindication of topical management the systemic hedgehog inhibitors should be taken into consideration.

Topical imiquimod 5% cream, a toll-like receptor 7 (TLR7) agonist, has emerged as a promising therapeutic option in this context. It activates innate and adaptive immune responses, stimulating the production of cytokines such as interferon-α and interleukin-12, thereby promoting antitumor activity [9,10]. Initially approved for superficial BCC, actinic keratosis, and genital warts [11], its application in nodular BCCs and periocular lesions has gained traction in recent years due to its immunomodulatory mechanism, low systemic absorption, and favorable cosmetic outcomes in this surgically challenging area [12]. Despite promising results (10% cumulative recurrence and 84% efficacy of treatment superficial and nodular BCCs with a 3 year follow-up period [7,13]), the ophthalmic safety profile of imiquimod, its long-term efficacy, and the risk of ocular surface irritation remain areas of concern [14].

This systematic review aims to critically evaluate the clinical efficacy, safety, patient-centered outcomes, and recurrence rates associated with imiquimod 5% cream for periocular BCC (superficial and nodular subtypes). We examine its role both as a primary treatment and as an adjunct and highlight methodological gaps to guide future research.

## 2. Materials and Methods

### 2.1. Registration and Reporting

A systematic review was carried out in accordance with the Preferred Reporting Items for Systematic Reviews and Meta-Analyses (PRISMA) guidelines [15]. The review protocol was registered with PROSPERO prior to the start of the screening process under number CRD420251018757, available from https://www.crd.york.ac.uk/PROSPERO/view/CRD420251018757 (accessed on 12 June 2025).

### 2.2. Screening and Data Extraction

The search to collect relevant manuscripts was performed in PubMed, Web of Science, and Scopus up to 12 June 2025. The search strategy, based on a combination of relevant medical subheadings (MeSHs), text words, and word variants for “topical”, “therapy”, “5% imiquimod”, and “eyelid” were utilized. The PubMed database was searched using the following key words: ((“imiquimod”)AND (“eyelid” OR “periocular”)) [16]. The Web of Science and Scopus database were searched as well, using the same strategy.

Details regarding the search strategy for each database are presented in Appendix A. Google Scholar was used as an additional citation tracking resource to search for any further studies not identified from the systematic search [17]. Two authors (L. K.-W. and P. S.) reviewed titles and abstracts independently. Full texts of papers were subsequently examined independently by both authors to determine if studies were eligible for inclusion in the review. Any uncertainty or disagreement about articles meeting the inclusion criteria were resolved after discussions among all authors to reach a consensus [18].

### 2.3. Types of Studies and Eligibility Criteria

The search results were systematically screened for inclusion based on pre-defined inclusion and exclusion criteria. Studies meeting the following inclusion criteria were included in this systematic review:
**1.** Population: Patients with histologically confirmed eyelid tumors (benign, premalignant, or malignant).**2.** Intervention: Studies evaluating imiquimod cream as a primary or adjunctive treatment.**3.** Comparison: Studies comparing imiquimod to other standard treatments (surgery, cryotherapy, radiation, PDT, 5-FU, corticosteroids, etc.).**4.** Outcomes: Must report treatment efficacy, recurrence rates, safety, or patient satisfaction.**5.** Study Designs:
(a)Randomized controlled trials (RCTs),(b)Observational studies (cohort, case–control, or cross-sectional studies).**6.** Language and Publication Status:
(a)Studies published in English,(b)Full-text available.

The exclusion criteria were as follows:**1.** Population:
(a)Studies on non-eyelid tumors (e.g., tumors of the face, scalp, or body),(b)Studies on non-tumor eyelid conditions (e.g., chalazion, blepharitis),(c)Studies on animal models or in vitro experiments.**2.** Intervention: Studies that do not use imiquimod cream.**3.** Comparison: Studies without a comparator treatment or only descriptive case reports.**4.** Outcomes: Studies that do not report relevant clinical outcomes.**5.** Study Types: Narrative reviews, expert opinions, or letters to the editor without primary data, case reports, and case series (with <5 patients).**6.** Language and Publication Status:
(a)Non-English publications without an available translation,(b)Unpublished or non-peer-reviewed studies.

Details regarding inclusion and exclusion processes are outlined in Figure 1 [19,20].

### 2.4. Assessment of the Level of Evidence

The level of evidence was assessed for each study using the approach proposed by the Grading of Recommendations, Assessment, Development and Evaluation (GRADE) Working Group. If serious issues were identified in risk of bias, inconsistency, indirectness, imprecision, or publication bias, the evidence level was downgraded accordingly. Therefore, the quality of evidence was divided into four levels (very low, low, moderate, and high) to indicate the level of confidence in the accuracy of the effect estimates.

The methodological quality of the included studies was evaluated according to the Cochrane collaboration’s tool for assessing the risk of bias for randomized clinical trials (RCTs) and observational studies [21,22,23,24].

### 2.5. Statistical (Data-Synthesis) Methods

In accordance with PRISMA 2020 guidelines, the study selection process was documented using a PRISMA flow diagram (Figure 1). Outcome data from each included study were systematically extracted. These data were subsequently presented in structured tables to facilitate the comparison of key findings [25].

Given the substantial heterogeneity across the included studies in design, interventions, outcome measures, and follow-up durations, a quantitative meta-analysis was not feasible. Consequently, a qualitative (narrative) synthesis of the findings was conducted. Pooled clearance rates were descriptive, and there were calculated using Excel without statistical weighting or heterogeneity testing [26].

## 3. Results

### 3.1. Study Selection

The database search yielded 51 records. After the removal of 2 duplicates and screening of titles/abstracts, 16 full-text articles were assessed for eligibility. Seven studies met all inclusion criteria and were included in the systematic review (Figure 1) [27]. These seven studies encompassed a total of 141 patients with 152 periocular BCC lesions. No additional eligible reports were identified through manual reference checks of pertinent articles [28].

### 3.2. Characteristics of Included Studies

The seven included studies were published between 2010 and 2022 and were conducted in five countries (Brazil, Spain, Greece, Australia, and India). The study designs comprised one randomized controlled trial, one nonrandomized comparative cohort, and five single-arm case series (prospective or retrospective). Sample sizes ranged from 8 to 47 patients. All studies investigated topical 5% imiquimod cream for periocular (eyelid or periocular skin) BCC, applied once daily, five times per week in most protocols. Treatment duration varied from 6 weeks to 16 weeks, with one study employing two cycles of imiquimod in combination with cryotherapy (termed “immunocryosurgery”). In one trial, imiquimod was compared to primary radiotherapy, and in one retrospective study it was compared to topical 5-fluorouracil. Follow-up periods for outcome assessment ranged from about 3 months in some reports to up to 5 years in others. The BCC lesions treated were generally histologically confirmed and predominantly of the nodular subtype; lesion size varied widely (from a few millimeters to as large as 40 mm in diameter in the immunocryosurgery series). Notably, none of the included studies provided a direct head-to-head comparison between imiquimod and surgical excision (the standard of care for BCC) [6,29,30,31,32,33,34,35]. As most studies did not report results separately for superficial vs. nodular BCC, stratified analysis by subtype was not possible. This limitation is addressed in the discussion.

### 3.3. Risk of Bias and Evidence Certainty

Using the Cochrane risk-of-bias tools, the single RCT was judged to have a relatively low risk of bias (with some concerns in minor domains), whereas the observational studies showed moderate-to-serious risk of bias due to their uncontrolled designs, heterogeneous methods, and lack of blinding [22,23]. In particular, there was variability in how outcomes were determined; some studies confirmed tumor clearance with histopathology, while others relied on clinical inspection alone, potentially allowing undetected residual tumor [29,30,31,32,33,34,35]. One cohort study mitigated this by excising the treatment site in all patients to verify clearance, but most studies used less rigorous assessments (e.g., limited post-treatment biopsy or purely clinical follow-up). In addition, the included studies were small and mostly positive, so the possibility of publication bias cannot be excluded (Figure 2).

The certainty of evidence was evaluated with GRADE criteria (Table 1) [23]. For the primary outcome of tumor clearance, the evidence was rated to have moderate certainty when clearance was confirmed by biopsy, but only rated to have very low certainty for clearance based on clinical exam without mandatory biopsy (downgraded due to risk of bias, inconsistency, and imprecision) [29,30,31,32,33,34,35]. The evidence for local tumor control over the longer term (recurrence-free survival) was low, reflecting the limited sample sizes and follow-up [30,31,33]. Evidence for safety outcomes was mixed, with moderate certainty for the predictably common local skin reactions, but low certainty for less frequent ophthalmic complications (owing to variability in reporting across studies) [29,30,31,32,33,34,35]. Evidence for cosmetic outcomes and patient-reported measures was very limited (very low certainty), as these outcomes were reported in only one randomized trial and one observational study using non-validated scales [30,35].

### 3.4. Primary Outcomes (Efficacy)

All included studies evaluated tumor clearance as a primary efficacy outcome, either by clinical examination or histologic confirmation [29,30,31,32,33,34,35] (Table 2). Reported clearance rates after imiquimod therapy were generally high, although definitions and time points varied. In a prospective case series by García-Martín et al. (2010), 100% of treated periocular BCCs (15/15) achieved complete histopathologic remission within 3 months of starting imiquimod [29]. Similarly, the subsequent randomized trial by García-Martín et al. (2011) found that 100% of lesions in both the imiquimod arm (15/15) and the radiotherapy arm (12/12) were free of tumor on biopsy 6 weeks after treatment, demonstrating non-inferiority of imiquimod to radiotherapy for short-term clearance [30]. Carneiro et al. (2010) reported an 80% clearance rate (8 of 10 lesions) confirmed by biopsy after 10–16 weeks of imiquimod, with no clinical recurrences during a mean follow-up of ~12 months [31]. In a larger nonrandomized cohort, de Macedo et al. (2015) observed 75% overall clearance (18/24 lesions) when all patients underwent surgical excision of the treatment area after imiquimod therapy; notably, clearance was 100% for tumors < 10 mm in diameter versus 82% for tumors > 10 mm over a follow-up period of 3 years [32].

The combination imiquimod-plus-cryotherapy approach by Gaitanis et al. yielded a high long-term success rate: 88% of periocular BCC lesions (14/16) were reported as completely “cleared” with sustained clinical remission through follow-ups up to 5 years [33]. In the only head-to-head comparison of topical treatments, Singh et al. (2022) treated 16 patients having extensive (“complex”) periocular BCC with imiquimod and 14 patients with 5-fluorouracil [35]. That retrospective study documented complete clinical tumor resolution in 10 of 16 imiquimod cases (62.5%) and 8 of 14 5-FU cases (~57%), with all responders remaining tumor-free at a median of 12 months post-therapy. Across these studies, tumor recurrence after initial clearance was uncommon within the reported follow-up intervals. No relapses were noted in the small RCT over 24 months of observation [30], and other series similarly reported durable local control in those lesions that responded to imiquimod [31,32,33]. However, two studies that relied on clinical judgment alone for determining clearance did not perform routine biopsy of the post-treatment site, which could underestimate the true residual tumor rate [34,35]. Overall, the efficacy data suggest that imiquimod 5% monotherapy achieves high initial clearance rates (approximately 75–100% in most series, depending on tumor size and study method) and can sustain long-term remission in the majority of responding lesions over 1–5 years of follow-up [29,30,31,32,33,34,35].

### 3.5. Secondary Outcomes

**Adverse events:** All studies reported on local and ocular safety outcomes (Table 3). As expected for an immune response modifier, local inflammatory reactions at the application site were almost universal. The treated periocular skin typically developed some degree of erythema, edema, erosions, crusting, and/or irritation in the course of therapy. In fact, nearly every patient experienced these local side effects (approximately ≥88% of lesions were noted to have erythema, crusting, or ulceration). These reactions were generally classified as mild-to-moderate in severity and self-limited, tending to resolve promptly with treatment interruptions or upon completion of the imiquimod course [29,30,31,32,33,34,35]. No cases of serious tissue damage from the cream were reported. Nonetheless, the intensity of local inflammation did impact tolerability for some patients; for example, in one series almost half of the participants described the local reaction severity as “poor tolerability,” although all of the patients were ultimately able to complete therapy [29].

**Ophthalmic complications** (affecting the conjunctiva, cornea, or ocular tissues) were reported with variable frequency, ranging from 0% to 100% of patients across different studies, owing to differences in how diligently such events were assessed [31,32,33,34,35]. The largest dedicated safety study by Cannon et al. with 47 patients involved noted that about one-third of patients (32%) developed mild conjunctivitis during imiquimod treatment [34]. Other ocular-region side effects in that cohort included reports of stinging or excessive tearing (13% of patients) and rare instances (<5%) of superficial keratitis or preseptal cellulitis. Importantly, no serious ocular injury (such as vision loss or persistent corneal damage) was observed [34]. All eye-related side effects were transient and resolved either spontaneously after drug cessation or with brief topical therapies, without any long-term sequelae. Consistently, across the seven studies there were no systemic adverse events attributable to topical imiquimod. Only a single patient (in García-Martín et al. 2010) reported mild systemic symptoms (fatigue/malaise), and no systemic illnesses or laboratory abnormalities were linked to treatment in any study [29,30,31,32,33,34,35].

**Cosmetic outcomes:** Cosmetic results and patient-reported outcomes were documented in a subset of studies. The randomized trial comparing imiquimod to radiotherapy found that imiquimod produced superior cosmetic outcomes—treated lesions healed with less scarring and better periocular cosmesis than those treated with radiotherapy [30]. This aligns with qualitative observations from the case series, which generally described favorable cosmetic results after imiquimod therapy, such as well-healed skin with minimal scarring or pigmentary change in the periocular area [29,31,32,33]. Patients appeared satisfied with the post-treatment appearance in most reports, although formal cosmetic grading scales were not uniformly used [30,35]. No study in this review employed a validated quality-of-life instrument, but the available descriptions suggest that imiquimod’s tissue-sparing approach yielded good functional and aesthetic outcomes in this delicate anatomical region, provided patients tolerated the temporary inflammatory reaction during treatment.

## 4. Discussion

This review synthesizes current evidence on the use of imiquimod 5% cream for the treatment of periocular basal cell carcinoma (BCC). The results indicate that imiquimod is a potentially effective, well-tolerated, and cosmetically favorable non-invasive treatment, particularly for patients with superficial or nodular BCC in anatomically sensitive regions such as the eyelid and medial canthus [6,8,29,30,32].

### 4.1. Efficacy and Indications

Surgical excision, particularly Mohs micrographic surgery, remains the gold standard for periocular BCCs, with established clearance rates in the range of 95% to 99% in the published literature [7]. Surgical techniques, especially those involving intraoperative margin control, are associated with more predictable long-term outcomes and lower recurrence rates.

In the reviewed studies, reported clearance rates were generally high, ranging from 75% to 100% [29,30,31,32,33,34,35]. These outcomes varied based on factors such as lesion size, anatomical location, treatment duration, and follow-up interval. Lesions smaller than 10 mm showed better outcomes, while larger or recurrent lesions had slightly lower clearance rates. These outcomes are notable considering the challenging location of the tumors, where surgical excision can risk functional and cosmetic compromise. The only randomized controlled trial confirmed that imiquimod is similarly effective as radiotherapy, with superior aesthetic results, supporting its role as a first-line of non-surgical therapy in selected cases [30] (Table 4).

### 4.2. Determinants of Response

Lesion size emerged as the most consistent predictor of durable clearance. In the prospective cohort from São Paulo, histological success at >3 years was 100% for tumors < 10 mm versus 82% for larger nodules [32]. Similar size dependent efficacy was evident in the Australian and Indian series, where complex lesions extending over >50% of lid length or measuring >20 mm required prolonged application and showed higher partial response rates [34,35]. Conversely, histological subtype (superficial vs. nodular), prior surgical relapse and metatypical differentiation did not consistently predict failure—provided that treatment was continued until a brisk inflammatory reaction was achieved.

However, while the results are promising, they should be interpreted with caution due to study limitations. Assessment methods varied markedly across studies, affecting the comparability of treatment outcomes for periocular BCC. Histological clearance was not uniformly confirmed. Studies using histological confirmation detected residual microscopic tumors undetectable by clinical inspection, while those relying solely on clinical criteria were at risk of overestimating cure rates as apparent healing may merely reflect re-epithelialization over persistent tumor nests rather than true eradication of malignancy [29,31,32,33,34,35].

Variation in biopsy techniques further complicated comparisons. Limited sampling methods, such as punch biopsies, could miss focal persistence, thereby falsely elevating clearance rates [29]. This limitation was mitigated in studies such as de Macedo at al., where all previously treated tissue was excised and subjected to histologic examination, allowing for more definitive conclusions regarding treatment success [32]. The timing of post-treatment assessment also varied substantially. Early biopsies (e.g., 6 weeks) may be confounded by inflammation, while delayed assessments risk missing early recurrence [30].

Many studies lacked control groups, randomization, or standardized protocols. Larger, well-designed trials are needed to validate efficacy across BCC subtypes, particularly nodular and infiltrative variants which may be less responsive to topical therapy alone.

Overall, “clearance” was inconsistently defined. To avoid overestimating efficacy, a synthesis of outcomes should stratify by verification method and follow-up duration, recognizing that studies using clinical inspection alone apply fundamentally lower thresholds for success.

### 4.3. Safety and Tolerability

Imiquimod was generally well tolerated. Most adverse effects were local, predictable, and transient, such as erythema, irritation, and periorbital edema in ≥70% of periocular applications. These inflammatory responses often reflect therapeutic activity and typically resolve post-treatment. Ophthalmic side effects, such as conjunctival irritation and blepharitis, were reversible in all cases and did not result in permanent ocular morbidity [34]. However, treatment near the eye should be performed with caution, under close monitoring, and with patient education regarding proper application.

The spectrum and intensity of side effects correlate with frequency of application. Daily regimens (5 times a week) triggered higher discontinuation rates in the Spanish case series (7/15 rated tolerability “bad”), whereas the 3 times a week schedule used by Cannon et al. was better tolerated with equivalent lesion resolution [34]. Liberal use of lubricating gels, protective eye shields during nocturnal weeping, and patient education on meticulous lid margin avoidance remain critical adjuncts.

It is worth mentioning that some of patients can have very subtle or even ephemeral symptoms during topical treatment with imiquimod, suggesting that immunological response is inadequate and other treatment modalities should be involved. Thus, the local side effects could be considered as a predictor of the efficacy of the treatment. However, none of the studies explored this relationship.

Importantly, no serious systemic side effects were reported, even with periocular application [29,30,31,32,33,34,35]. This favorable safety profile supports its use in elderly and frail patients who may not tolerate surgery or radiotherapy. Treatment with use of imiquimod cream was slightly worse tolerated than radiotherapy because of local irritation, though functional final outcomes were superior [30].

While some patients did experience side effects, the overall tolerance to imiquimod was relatively good. This finding is noteworthy as it suggests that patients may tolerate topical immunotherapy better than anticipated, especially in sensitive areas like eyelids.

### 4.4. Cosmesis

Although inconsistently quantified, cosmesis was uniformly favorable. The RCT reported better lid pliability and fewer lash losses with imiquimod than with radiotherapy [30]. Photographic documentation from long term cohorts shows negligible cicatricial distortion even after intense inflammatory phases, and any hypertrophic scarring subsided over time.

Given the eyelid location, cosmetic outcome is crucial. All studies with long-term follow-up noted excellent cosmesis with imiquimod—an important finding since surgical excision (especially near the canthus) often requires reconstruction that can alter eyelid contour [30,31,32]. Imiquimod’s ability to clear tumors while preserving normal tissue results in negligible scaring for most patients.

### 4.5. Patient-Centered Outcomes

Although formal quality-of-life (QoL) metrics were largely absent, some studies suggested high patient satisfaction and good cosmetic outcomes, particularly when compared to more invasive alternatives [29,30,31,32,33,34,35]. Patients frequently reported satisfaction with the treatment course and clinicians noted the preservation of lid contour and function, which can be compromised following excision and reconstruction. While the absolute clearance rates with imiquimod may be modestly lower than those achieved with surgery, its benefits in terms of tissue preservation, patient tolerability, and suitability in high-risk populations underscore its clinical relevance as a non-surgical alternative in selected cases.

Not all patients found the treatment easy—in one series nearly half rated tolerability as ‘poor’ due to local side effects, underscoring that while non-surgical the therapy is not trivial [29]. Nonetheless, the overall acceptance was good in other reports, and patients valued avoiding surgery [31,32,33,34,35].

There is a need for future studies to incorporate patient-reported outcome measures (PROMs), which are essential for evaluating treatment impact in periocular disease, where both function and appearance are paramount.

### 4.6. Combination Therapy and Emerging Approaches

The combination of imiquimod with cryosurgery (immunocryosurgery) represents a promising approach that may enhance efficacy while maintaining minimal invasiveness [33]. This strategy could be particularly useful for nodular or recurrent tumors, although evidence is currently limited to small case series. Future comparative trials are needed to assess the additive benefit of combined modalities versus monotherapy. Emerging evidence also supports mild mechanical debulking or adjuvant cryotherapy to enhance penetration. These synergistic protocols may compensate for the diminished drug diffusion seen in thick nodules while avoiding the functional deficits of radical surgery.

Although evidence from other contexts suggests that combining imiquimod with fractional laser or microneedling can enhance efficacy, none of the included studies examined such combinations in periocular BCCs. Fractional laser pre-treatment can enhance topical drug penetration, potentially increasing efficacy—future exploration of this in periocular BCC would be valuable.

Moreover, none of the studies considered the neoadjuvant use of imiquimod in order to decrease the volume of the tumor as one of the options in locally advanced or aggressive subtypes of BCCs. Using imiquimod pre-operatively could reduce tumor size, possibly allowing smaller surgical excisions; this strategy is worth investigating for locally advanced BCC to improve surgical outcomes.

### 4.7. Follow-Up and Recurrence Risk

The duration of follow-up played a critical role in evaluating long-term efficacy. Given that recurrence of periocular BCC often peaks within the first two to three years post-treatment, studies with only short-term follow-up—such as de Macedo et al., which included histologic clearance assessments at 6–12 weeks—are not directly comparable to studies with extended clinical surveillance, including serial evaluations over two years [30,31,32,33]. Taking it all into consideration, follow-up visits should be performed every 6 months for at least 3 years after treatment, regardless of BCC subtype.

Since most reviewed studies were performed in ophthalmology departments, the following methods were used to assess recurrence: clinical examination (including slit-lamp in ophthalmologic settings), standardized photography to track lesion appearance, and histopathological confirmation via repeat biopsy, particularly 3 months post-treatment or upon suspicion of relapse [29,30,31,32,33,34,35].

According to the rising trend in the use of different imaging techniques in medicine, more advanced methods should be applied in assessment of efficacy and recurrence rates (e.g., reflectance confocal microscopy, line field confocal microscopy, or high-frequency ultrasound).

Consistent use of dermoscopy, imaging, and biopsy where indicated is essential in routine clinical practice to ensure complete response and monitor for recurrence. Long-term surveillance strategies should be standardized in future protocols.

### 4.8. Limitations of the Evidence Base

The quality of evidence remains limited. Most studies were observational, with small sample sizes (<30 lesions)—impacting statistical power, potential selection bias, and limited follow-up. Only one trial employed randomization [29,30,31,32,33,34,35]. There was also significant heterogeneity in lesion types, treatment regimens (e.g., dosing frequency, duration, and protective measures), and outcome measures. The absence of randomized comparisons with standard surgical techniques or Mohs micrographic surgery limits the generalizability of findings. Most studies relied on general observations and lacked validated tools to measure patient-reported discomfort or the impact on daily life and emotional well-being. Moreover, few studies stratified outcomes by tumor characteristics, such as size, depth, or histological subtype. Follow-up seldom exceeded 36 months, yet periocular BCCs may recur even after 5 years [5,30,32].

It is also possible that small studies with negative or poor outcomes were not published (publication bias), given the generally favorable results in the literature [24].

Future trials should therefore adopt uniform, histologically confirmed endpoints at ≥12 weeks; Kaplan–Meier recurrence analysis with ≥5 year surveillance; patient reported outcome measures (PROMs) for cosmesis and quality-of-life; and stratified comparison against Mohs surgery and hedgehog pathway inhibitors.

### 4.9. Position of Imiquimod in the Periocular Superficial or Nodular BCC Treatment Algorithm

For frail, multimorbid, or surgery averse patients—and for tumors abutting the lacrimal drainage apparatus or free lid margin—imiquimod therefore offers a unique balance of oncological control and lid preservation [29,30,31,32,33,34,35]. Taking everything into consideration, the data support the following risk-adapted therapeutic approach:1.Low risk tumors (<10 mm, primary, superficial/nodular, away from punctum): Topical imiquimod 5% three to five times weekly for 6–10 weeks.2.Intermediate risk lesions (10–20 mm, nodular, proximity to lacrimal drainage or prior surgery): Consider immunocryosurgery or imiquimod neoadjuvant therapy to debulk prior to limited excision [33].3.High risk tumors (>20 mm, infiltrative/morpheaform, orbital invasion): Primary Mohs surgery or, where contraindicated, systemic hedgehog inhibitors [7].

Importantly, biopsy-verified clearance is essential for all nonsurgical cases, and patients should be counseled that a robust inflammatory reaction and transient ocular irritation are expected with imiquimod (a sign of its activity).

## 5. Conclusions

Imiquimod 5% achieves a durable cure in roughly four out of five cases and excellent cosmesis for selected periocular BCCs (superficial and nodular subtypes), with predictable, reversible local irritation as the main drawback. While current evidence supports its use when surgery or radiotherapy is undesirable, multicenter RCTs with ≥3 year follow-up are required to more definitively establish its long-term efficacy and optimal role in periocular BCC management [29,30,31,32,33,34,35].

## Figures and Tables

**Figure 1 cancers-17-02111-f001:**
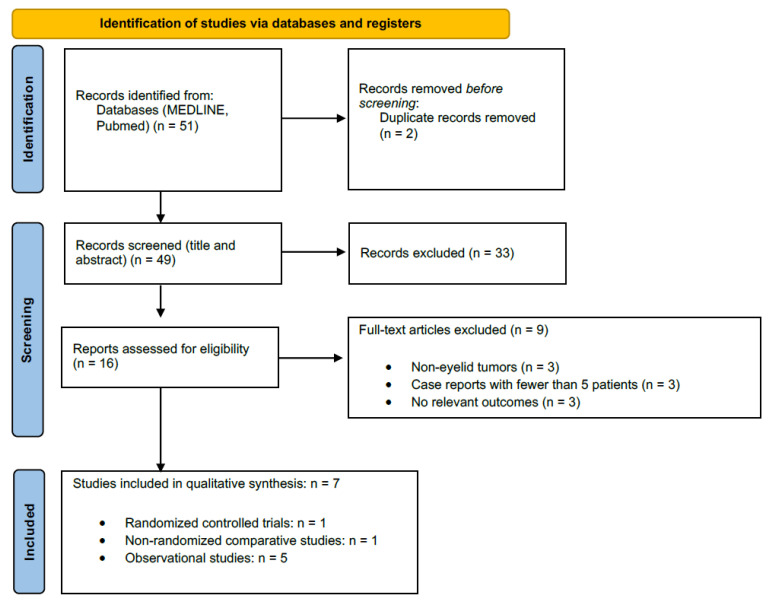
PRISMA flow diagram describing the selection of studies included in the systematic review.

**Figure 2 cancers-17-02111-f002:**
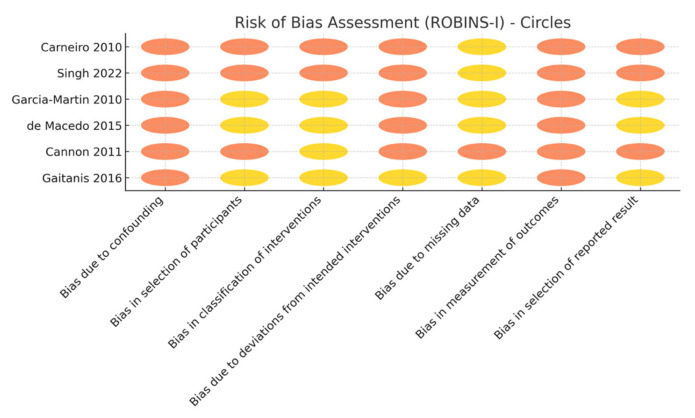
Risk of bias assessment of the observational studies (yellow—moderate risk, orange—high risk).

**Table 1 cancers-17-02111-t001:** GRADE evidence profile for key outcomes of periocular imiquimod 5% cream.

Outcome	No. Studies/Lesions (Patients)	Study Design(s)	Risk of Bias	Inconsistency	Indirectness	Imprecision	Publication Bias	Overall Certainty	Downgrade Rationale
	Four studies/64 lesions (56 pts)	A total of 1 RCT + 3 prospective cohorts	Not serious (RoB 2: low concern; cohorts had prospective protocols)	Not serious	Not serious (direct periocular BCC)	Serious (total n < 100; 95% CI 86–97%)	Suspected (small positive trials)	MODERATE	–1 imprecision
Clinical-only clearance	Three studies/79 lesions (58 pts)	Retrospective or uncontrolled series	Serious (no controls and subjective outcome)	Serious (protocols differ)	Serious (indirect surrogate for cure)	Serious (wide CI 57–81%)	Likely	VERY LOW	–1 RoB, –1 indirectness, and –1 imprecision
Local periocular skin AEs (erythema, crusting, ulceration)	Six studies/122 pts	Retrospective or uncontrolled series	Serious (selective AE reporting in two studies)	Not serious (all report ≥ 60% events)	Not serious	Serious (denominators unclear in two studies)	Possible	MODERATE	–1 imprecision
Ophthalmic AEs(conjunctivitis, keratitis, cellulitis)	Four studies/94 pts	Retrospective or uncontrolled series	Serious (non-masked AE ascertainment)	Serious (rates 0–100%)	Not serious	Serious (event numbers small)	Possible	LOW	–1 RoB and –1 inconsistency/imprecision
Cosmetic and PROM outcomes	Two studies/42 pts	RCT + retrospective	Serious (non-validated scales and reporting bias)	Serious (different metrics)	Serious (indirect surrogate of QoL)	Serious (n < 50)	Likely	VERY LOW	–1 RoB, –1 indirectness, and –1 imprecision

Abbreviations: pts, patients; RCT, randomized controlled trial; AE, adverse events; QoL, quali-ty-of-life; CI, confidence interval; RoB, risk of bias.

**Table 2 cancers-17-02111-t002:** “Clearance” definition and measurement.

Study (Design)	N Lesions	Authors’ Term for Success	How Success Was Confirmed	Timing of Assessment
Carneiro 2010 [31]—prospective case series	10	“clinical and histological resolution”	Slit-lamp/photographic inspection plus repeat 2 mm punch biopsy of every treated lesion	A total of 12 weeks after the end of therapy
García-Martín 2010 [29]—prospective case series	15	“histopathological remission” (sustained clinical remission reported separately)	Mandatory post-treatment biopsy of the original tumor bed	≤3 months after starting imiquimod; followed clinically to 24–28 months
García-Martín 2011 [30]—randomized comparison IMQ vs. radiotherapy	15 IMQ, 12 RT	“histopathological remission”	Biopsy of the treated site 6 weeks after completing the 6 week course	Histology at 6 weeks; clinical FU 24 months
de Macedo 2015 [32]—nonrandomized cohort	24	“75% histological clearance”	Post-treatment biopsy	Not stated
Singh 2022 [35]—retrospective series (complex BCC)	16 IMQ, 14 5-FU	“complete clinical tumor resolution”	External inspection (no routine biopsy); recurrence judged clinically during ≥12 month follow-up	Median 12–16.5 weeks after start; clinical FU ≥ 12 months
Gaitanis 2016 [33]—retrospective immunocryosurgery series	16	“cleared”/“sustained clinical remission”	Clinical examination only (no biopsy reported)	End of each 5- or 10-week cycle; clinical FU 3–60 months
Cannon 2010 [34] —safety series	47	Outcome not primary endpoint (34/47 “clinical resolution”)	Clinical assessment; no histology	Mean 16 weeks after therapy

**Table 3 cancers-17-02111-t003:** Adverse event profile of periocular imiquimod.

Study (Year)	Treated pts/Lesions	Local Periocular Skin Events	Ophthalmic Events (Conjunctiva/Cornea/Orbit)	Systemic Events	Notes on Data Quality
Carneiro (2010) [31]	8/10	Hyperaemia, crusting, ulceration, and bleeding reported in all patients (8/8)	Keratitis punctata ± allergic conjunctivitis (number not stated)	0	AEs described qualitatively; counts not provided.
García-Martín (2010) [29]	15/15	Local inflammatory reactions in 15/15; 7 judged “bad tolerability”	None specifically ocular	0	Ocular side-effects not itemized.
García-Martín (2011) [30]	15/15	Eyelid discomfort when blinking 9/15	Conjunctival irritation 2/15	0	All reactions resolved after therapy.
de Macedo (2015) [32]	24/24	Mild local reactions (number not given)	NR	0	Only statement that AEs were “mild”; no counts.
Singh (2022) [35]	16/16	Periocular erythema; skin depigmentation 16/16	Chemical conjunctivitis 16/16	0	All ocular events reversible.
Gaitanis (2016) [33]	16/16	Limited morbidity– no cutaneous complications quantified	0	0	Authors state “minimal” AEs; none detailed.
Cannon (2010) [34]	47/47	Application-site erythema 47/47; vesicles/ulceration 3	Conjunctivitis 15; ocular stinging 6; keratitis 1; delayed conjunctivitis 3; pre-septal cellulitis 2 → ≥24/47 unique pts	0	AEs listed individually; overlap between events possible.

**Table 4 cancers-17-02111-t004:** Comparison of imiquimod cream with other treatments for eyelid BCC.

Treatment	Efficacy Rate	Key Advantages	Key Limitations	Citation
Imiquimod 5% Cream	76–88% (varies by study and follow-up)	Non-invasive; suitable for delicate eyelid areas; and cosmetic preservation	Local skin reactions common; variable response; and less effective in nodular/infiltrative types	Singh et al., 2022 [35]; Garcia-Martin et al., 2010 [29]
Radiotherapy	~90–95%	Useful in elderly or inoperable patients; and organ-sparing	Multiple sessions; and risk of long-term skin atrophy or pigmentary changes	Garcia-Martin et al., 2011 [30]
Imiquimod + Cryotherapy	~90%	Potential synergistic effect; and shortened treatment duration	Limited studies; and risk of cryotherapy complications	Gaitanis et al., 2016 [33]
Mohs Micrographic Surgery	97–99%	Tissue-sparing; maximal margin control; and best for recurrent/infiltrative tumors	Limited availability; expensive; and longer procedural time	Standard clinical reference (not from included SR studies)

## Data Availability

Data are available on request from the corresponding author.

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
