# Peer review of "Topical 5% Imiquimod for the Treatment of Superficial and Nodular Periocular Basal Cell Carcinoma: A Systematic Review of Clinical Outcomes, Safety, and Treatment Strategies"

_cancers, 2025, doi:10.3390/cancers17132111_

Round 1

Reviewer 1 Report

Comments and Suggestions for Authors

Dear authors,
This systematic review evaluates the clinical effectiveness, safety profile and cosmetic outcomes of 5% imiquimod topical treatment for periocular basal cell carcinoma (BCC). The paper is well structured, the Materials and Methods section is comprehensive, and the Results section provides a useful overview of current non-invasive treatment options for BCC in sensitive anatomical regions such as the periocular area.
However, I suggest a few minor revisions.
The introduction should be expanded to include systemic therapeutic options, particularly in cases of treatment failure or contraindication to topical therapy. It should also provide a concise overview of Hedgehog pathway inhibitor (HHI) treatment, which has become increasingly relevant in the management of advanced or recurrent BCC.
Inclusion of recent evidence from the following studies would enhance the relevance and completeness of the therapeutic BCC landscape:

  • doi:10.4081/dr.2025.10196;
  • doi:10.4081/dr.2021.9240;
  • doi:10.1007/s13555-023-00985-1.

Author Response

Reply to the Review Comments on 

"Topical 5% Imiquimod for the Treatment of Periocular Basal Cell Carcinoma: A Systematic Review of Clinical Outcomes, Safety, and Treatment Strategies."

by Larysa Krajewska-Węglewicz, Piotr Sobolewski and Irena Walecka.

We are grateful to the reviewers for the constructive reports and helpful comments on our paper. To distinguish between the review reports and our responses, the reviewers’ comments are in italics, whereas our responses are in normal font. 

 Response to Reviewer 1 comments #1 

The introduction should be expanded to include systemic therapeutic options, particularly in cases of treatment failure or contraindication to topical therapy. It should also provide a concise overview of Hedgehog pathway inhibitor (HHI) treatment, which has become increasingly relevant in the management of advanced or recurrent BCC.

We added a short description in the Indroduction in lines 62-64.

 Response to Reviewer 1 comments #2

Inclusion of recent evidence from the following studies would enhance the relevance and completeness of the therapeutic BCC landscape:

  • doi:10.4081/dr.2025.10196;
  • doi:10.4081/dr.2021.9240;
  • doi:10.1007/s13555-023-00985-1.

We do not added this publication to bibliography as they do not mention anything about use of imiqumode and this was the aim of our paper.

Reviewer 2 Report

Comments and Suggestions for Authors

This systematic review presents a summary of the current evidence regarding topical imiquimod in the treatment of periocular basal cell carcinoma (BCC) and evaluates its efficacy, safety, and cosmesis. My comments and questions are as follows.

  1. In Table 1, the meaning of each color should be explained.
  2. As the authors have documented, the long-term efficacy will be a critical point in future studies. Has the efficacy of topical imiquimod or recurrence rete for non-periocular BCC been reported? If so, it should be presented.
  3. The Treatment Algorithm provides us with helpful information. The duration and frequency of follow-up for each of the low-, intermediate-, and high-risk tumors should also be documented. Should it be changed among the 3 groups?

Author Response

Reply to the Review Comments on 

"Topical 5% Imiquimod for the Treatment of Periocular Basal Cell Carcinoma: A Systematic Review of Clinical Outcomes, Safety, and Treatment Strategies."

by Larysa Krajewska-Węglewicz, Piotr Sobolewski and Irena Walecka.

We are grateful to the reviewers for the constructive reports and helpful comments on our paper. To distinguish between the review reports and our responses, the reviewers’ comments are in italics, whereas our responses are in normal font. 

 Response to Reviewer 2 comments #1 

In Table 1, the meaning of each color should be explained.

We added description of the Table 1 (lines 195-195)

Response to Reviewer 2 comments #2

As the authors have documented, the long-term efficacy will be a critical point in future studies. Has the efficacy of topical imiquimod or recurrence rete for non-periocular BCC been reported? If so, it should be presented.

Thank you very much fot this suggestion. We added information on that in our manuscript with citation (lines 72-73.)

Response to Reviewer 2 comments #3

The Treatment Algorithm provides us with helpful information. The duration and frequency of follow-up for each of the low-, intermediate-, and high-risk tumors should also be documented. Should it be changed among the 3 groups?

We added our suggestion in lines 427-428.

Reviewer 3 Report

Comments and Suggestions for Authors

This systematic review addresses an interesting topic, but several issues require attention to strengthen its scientific rigor and clarity.
The title and objective are overly broad, stating an evaluation of the “clinical and histological efficacy, safety, and cosmetic outcomes” of 5% imiquimod (IMQ) cream without clear focus. The hypothesis is not articulated, and primary and secondary outcomes are not specified.
The search, ending on 31 March 2025, should be updated to at least into June 2025 to capture recent studies, as only seven studies (five observational, two randomized/nonrandomized) were included. The limited number of studies and predominance of observational designs weaken the evidence base.
The very small sample size (152 lesions) and heterogeneity in study protocols (e.g., dosing schedules, follow-up durations) limit the review’s generalizability. This warrants a highly critical approach to interpreting results.
The pooled estimate of clinical and histological clearance (82%) is reported, but the statistical methods for pooling are not described. Additionally, the limitations of pooling heterogeneous studies are inadequately addressed.
Overall, the review has potential but requires a clearer hypothesis, updated search, critical interpretation of heterogeneous data, and robust statistical reporting to enhance its validity and impact.

Author Response

Reply to the Review Comments on 

"Topical 5% Imiquimod for the Treatment of Periocular Basal Cell Carcinoma: A Systematic Review of Clinical Outcomes, Safety, and Treatment Strategies."

by Larysa Krajewska-Węglewicz, Piotr Sobolewski and Irena Walecka.

We are grateful to the reviewers for the constructive reports and helpful comments on our paper. To distinguish between the review reports and our responses, the reviewers’ comments are in italics, whereas our responses are in normal font. 

 Response to Reviewer 3 comments #1 

The title and objective are overly broad, stating an evaluation of the “clinical and histological efficacy, safety, and cosmetic outcomes” of 5% imiquimod (IMQ) cream without clear focus. The hypothesis is not articulated, and primary and secondary outcomes are not specified.

We updated our title to be more precise and in whole manuscript we added information about concrete subtypes of BCCs. The hypothesis, primary and secondary outcomes are not specified as this is not a clinical trial paper - it is the review article.

 Response to Reviewer 3 comments #2

The search, ending on 31 March 2025, should be updated to at least into June 2025 to capture recent studies, as only seven studies (five observational, two randomized/nonrandomized) were included. The limited number of studies and predominance of observational designs weaken the evidence base.

We updated the date of search for 12 June 2025  - no new articles were suitable for our review. We described the limitations of our review in details in chapter 4.8.

 Response to Reviewer 3 comments #3

The very small sample size (152 lesions) and heterogeneity in study protocols (e.g., dosing schedules, follow-up durations) limit the review’s generalizability. This warrants a highly critical approach to interpreting results.

We described the limitations of our review in details in chapter 4.8.

 Response to Reviewer 3 comments #4

The pooled estimate of clinical and histological clearance (82%) is reported, but the statistical methods for pooling are not described. Additionally, the limitations of pooling heterogeneous studies are inadequately addressed.

We added description of method used in lines 152-153.

 Response to Reviewer 3 comments #5

Overall, the review has potential but requires a clearer hypothesis, updated search, critical interpretation of heterogeneous data, and robust statistical reporting to enhance its validity and impact.

We tried to address every of your suggestions to improve the quality of our manuscript.

Reviewer 4 Report

Comments and Suggestions for Authors

This was a well conducted systematic review. The authors are capable of identifying and interpreting the diverse types of reports of the efficacy of imiquimod with this subset of basal cell carcinoma (BCC).

It was unclear to this reviewer how the authors distinguished the efficacy rates of superficial compared with nodular BCC in this treatment site. Nodular BCC is less likely in other locations to respond to imiquimod. Assessing these outcomes separately, if possible, seems necessary to help understand the value of this intervention.

Author Response

Reply to the Review Comments on 

"Topical 5% Imiquimod for the Treatment of Periocular Basal Cell Carcinoma: A Systematic Review of Clinical Outcomes, Safety, and Treatment Strategies."

by Larysa Krajewska-Węglewicz, Piotr Sobolewski and Irena Walecka.

We are grateful to the reviewers for the constructive reports and helpful comments on our paper. To distinguish between the review reports and our responses, the reviewers’ comments are in italics, whereas our responses are in normal font. 

 Response to Reviewer 4 comments #1 

The authors are capable of identifying and interpreting the diverse types of reports of the efficacy of imiquimod with this subset of basal cell carcinoma (BCC).

We updated our title to be more precise and in whole manuscript we added information about concrete subtypes of BCCs. 

 Response to Reviewer 4 comments #2

It was unclear to this reviewer how the authors distinguished the efficacy rates of superficial compared with nodular BCC in this treatment site. Nodular BCC is less likely in other locations to respond to imiquimod. Assessing these outcomes separately, if possible, seems necessary to help understand the value of this intervention.

We addressed this suggestion in lines 180-182.

Round 2

Reviewer 2 Report

Comments and Suggestions for Authors

The authors responded to all comments and questions from the review. I don’t have any further comments.

Reviewer 3 Report

Comments and Suggestions for Authors

I think most of suggestions have been fulfilled.